# Serving Temperatures of Best-Selling Coffees in Two Segments of the Brazilian Food Service Industry Are “Very Hot”

**DOI:** 10.3390/foods9081047

**Published:** 2020-08-03

**Authors:** Ian C. C. Nóbrega, Igor H. L. Costa, Axel C. Macedo, Yuri M. Ishihara, Dirk W. Lachenmeier

**Affiliations:** 1Department of Food Engineering, Universidade Federal da Paraíba (UFPB), João Pessoa, Paraiba 58051-900, Brazil; igorhenr.98@gmail.com (I.H.L.C.); axeelmacedo@hotmail.com (A.C.M.); yuriufpb@yahoo.com.br (Y.M.I.); 2Chemisches und Veterinäruntersuchungsamt (CVUA) Karlsruhe, Weissenburger Str. 3, 76187 Karlsruhe, Germany

**Keywords:** coffee, temperature, risk, food service industry, Brazil

## Abstract

The International Agency for Research on Cancer has classified the consumption of “very hot” beverages (temperature >65 °C) as “probably carcinogenic to humans”, but there is no information regarding the serving temperature of Brazil’s most consumed hot beverage—coffee. The serving temperatures of best-selling coffee beverages in 50 low-cost food service establishments (LCFS) and 50 coffee shops (CS) were studied. The bestsellers in the LCFS were dominated by 50 mL shots of sweetened black coffee served in disposable polystyrene (PS) cups from thermos flasks. In the CS, 50 mL shots of freshly brewed espresso served in porcelain cups were the dominant beverage. The serving temperatures of all beverages were on average 90% and 68% above 65 °C in the LCFS and CS, respectively (P95 and median value of measurements: 77 and 70 °C, LCFS; 75 and 69 °C, CS). Furthermore, the cooling periods of hot water systems (50 mL at 75 °C and 69 °C in porcelain cups; 50 mL at 77 °C and 70 °C in PS cups) to 65 °C were investigated. When median temperatures of the best-selling coffees are considered, consumers should allow a minimum cooling time before drinking of about 2 min at both LCFS and CS.

## 1. Introduction

It is well established that Brazil is the world’s largest producer and exporter of coffee [1,2], but coffee consumption aspects in the country are less known. It has been reported that coffee amounts to 71% of total revenue in the hot drinks market (coffee, tea and cocoa) [3] and is the most commonly consumed food in Brazil after rice [4]. With regard to per capita intakes of the beverage in Brazil, however, data are scattered, limited and not as straight forward to obtain as coffee production and exports. One study, though, based on a 2008–2009 dietary survey, estimated a daily coffee intake of 163 mL per person in Brazil [5] while another, based on data from the Brazilian Coffee Industry Association, estimated the value in 2016 as 220 mL [6] (p. 65). However, these intake values taken alone are out of perspective. Since two key data—namely the annual coffee consumption in kg by regions/selected countries and corresponding populations by age groups—are available [7,8], it is possible to put estimates of Brazilian per capita daily intakes into perspective.

Assuming coffee consumption data from the International Coffee Organization [7] is in green coffee form, we applied a 1.19 conversion factor to roasted coffee (1.19 kg green coffee = 1.0 kg of roasted coffee) [9], then a coffee-to-water ratio of 1 to 12.5 (8 g roasted coffee/100 mL water) [10,11] (p. 25, p. 3) and distributed the values across populations aged 15 years and older [8] over 365 days. Taking these considerations into account, we estimate Brazilian per capita daily coffee intakes of 223 and 226 mL in 2015 and 2019, respectively. By applying the same approach to other areas, it can be stated with reasonable confidence that coffee intakes in Brazil were above all world regions in 2015 and 2019, for instance, Europe (144 and 153 mL), South America (140 and 140 mL) and Asia-Oceania (17 and 17 mL). Brazilian intakes are also higher than in countries such as the USA (169 and 177 mL), Japan (121 and 117 mL) and neighboring coffee producing Colombia (84 and 77 mL), but lower than Switzerland (251 and 262 mL) and Norway (343 and 348 mL) (see Appendix A).

In Brazil, retail prices of roasted coffee products reflect their quality certifications (in particular those issued by private coffee associations), such as “traditional” (lower price), “gourmet” and “especial“/specialty coffee (upper prices). Prices of coffee beverages in the Brazilian food service industry also vary with the segment. For instance, a 50 mL shot of espresso—which is typically brewed with “gourmet” or “especial” coffee—costs about BRL 6 in coffee shops; on the other hand, a 50 mL shot of filtered black coffee—commonly brewed with “traditional” coffee or even non-certified coffee—costs around BRL 1 in ordinary (low-cost) food service establishments. In the context of coffee consumption, these considerations are relevant because Brazil, an upper middle-income economy according to the World Bank [12], has a relatively low monthly household income per capita (BRL 1479 or USD 375 in 2019) [13] and 26.5% of its population live below the national poverty line income [14].

In 2016, the WHO’s International Agency for Research on Cancer (IARC) classified the consumption of “very hot” beverages (temperature > 65 °C) as “probably carcinogenic to humans” (IARC group 2A), with esophageal cancer particularly associated with the drink habit in human epidemiological studies [6,15]. IARC classifications indicate the weight of the evidence as to whether an agent is capable of causing cancer (technically called “hazard”) but it does not measure the likelihood that cancer will occur (technically called “risk”). Category 2A is used for an agent when there is limited evidence of carcinogenicity in humans and sufficient evidence of carcinogenicity in experimental animals [16]. It is important to highlight that 29 human epidemiological studies (on esophageal cancer and drinking very hot beverages) served as scientific evidence for the IARC classification in 2016, most of which related to the consumption of hot tea [6] (p. 451–465). Two prospective cohort studies published after the IARC classification, one by Yu et al. in 2018 [17] and another by Islami et al. in 2020 [18], again with tea, have further strengthened the association between consumption of very hot beverages and increased esophageal cancer risk; however, while the former study reported a positive association between hot tea drinking and esophageal cancer only among those with tobacco or heavy alcohol use, the latter found that the association was independent of these risk factors.

With regard to the risk of very hot beverages to humans, an important cancer risk assessment using the margin of exposure approach (considered the gold standard for assessing agents that are both genotoxic and carcinogenic) was published in 2018. The margin of exposure (MOE) is defined as the ratio between the point on the dose response curve which characterizes low but measurable harmful effects in animal studies (typically, the benchmark dose, or BMD, for a benchmark response of 10%) and the estimated human exposure to the agent, or simply put: MOE = BMD/exposure (clearly, the lower the MOE, the larger the risk for humans and a threshold of 10,000, or higher, is generally considered a low public health concern). The estimated MOE in the 2018 hot beverage study, using the BMD dose, was <1 (meaning that the human exposure had exceeded the dose that may cause a 10% cancer incidence in experimental animals) for individuals that drink as little as 60 mL of very hot beverages five times per week (calculated for a 60 kg person), suggesting very hot beverages pose a significant esophageal cancer risk even for the moderate drinking population [19].

According to Brazil’s National Cancer Institute (INCA), a specialized body of the Ministry of Health, the age-standardized incidence rates of esophageal cancer in Brazil (2020 estimates) are approximately seven and three cases in 100,000 men and women, respectively. Furthermore, esophageal cancer is the sixth most common cancer site among men in Brazil (excluding non-melanoma skin cancer) and projections for 2020 indicate 8690 total new cases. The habits listed by INCA as risk factors for the disease are alcohol consumption, smoking and drinking very hot beverages [20] (p. 52; p. 42).

Although drinking a cup of hot coffee is such a common habit in many countries, there are only a very limited number of investigations of the actual serving or dispensing temperatures of the beverage in places such as households, food service industry and workplace, but two publications are worth mentioning. One is a study published in 2007 on the recorded serving temperatures of 164 samples of black coffees in quick service restaurants in the USA, including well-known chains such as McDonald’s, Burger King, Wendy’s and KFC; the measured temperatures of all samples were on average 9 °C above the IARC threshold for very hot beverages (mean and median of all measurements: 74 °C; standard deviation: 13 °C) [21]. The other is a German study published in 2018 on the recorded serving temperatures of 356 coffee beverages in the food service industry and the dispensing temperatures of 110 coffees in private households; the measured temperatures were on average 10 °C above the IARC threshold temperature both in the household and in the food service industry (mean value of all measurements: 75 °C; standard deviation: 5 °C) [22]. These high serving temperatures are probably related to the commonly recommended temperature of ~93 °C for brewing coffee [23].

Interestingly, it has been reported by several studies that the most preferred or recommended drinking temperatures of black coffee are often well below the mean serving temperatures encountered in the two mentioned studied (~75 °C), for instance (decimals were rounded to the near whole number), 68 °C [24], 72 °C [25], 60 °C [26], 58 °C [27] and 63 °C [28].

Based on the previous considerations, the aim of this paper is to provide an initial set of data on the serving temperatures of best-selling coffee-based hot beverages (e.g., espresso, cappuccino, filtered black coffee, etc.) from two socio-economic segments of the Brazilian food service industry: coffee shops and low-cost food service establishments. The results are mainly discussed in terms of the IARC threshold temperature and in light of practical recommendations for coffee consumers.

## 2. Materials and Methods

### 2.1. Laboratory

Experiments described in Section 2.3 and Section 2.5 were conducted in the beverage lab of the Department of Food Engineering, UFPB University, Brazil. The laboratory is fully climatized with a controlled air temperature and its GPS coordinates are 7°08′32′′ S 34°50′58′′ W.

### 2.2. Field Work (Food Service Industry)

Experiments described in Section 2.4 were carried out between October and December 2019 in 100 food service establishments situated in João Pessoa, capital of the northeastern state of Paraíba, Brazil, which has a population of about 800,000. To obtain a reasonable representation of Brazilian coffee consumers from different socio-economic backgrounds, we sampled equal proportions of two segments of the food service establishments: coffee shops (CS) and low-cost food service establishments (LCFS). The selection process involved research using internet searches and site visits to identify potential establishments in a circular geographic area of approximately 250 km^2^ from the laboratory (radius = 9 km).

LCFS were predefined as having these main characteristics: lower-price (up to BRL 2.00 per cup of coffee), a limited offer of coffee-based beverages (typically, no more than 3 types), counter or table service (typically counter service only) and located in lower middle- or lower-income neighborhoods (the typical choice was establishments within or surrounding open air markets or farmers markets). On the other hand, CS were predefined as having a large number of coffee options on the menu (typically more than 10), table service, prices up to BRL 15.00 per cup and located in upper middle- or high-income neighborhoods. All 100 sampled establishments were outside the university campus.

### 2.3. Calibration Checking of Thermometers

Temperatures were measured using a digital food thermometer device “El Corte Inglés” (EAN:2401662264904) with a stainless steel sensor, heat-resistant cable, range 0 to +300 °C and LCD display (El Corte Inglés, Madrid, Spain).

We assessed the calibration of the digital thermometer using a Braun thermostatic bath model 18 BU coupled with a Thermomix BM-S (B. Braun Biotech International, Melsungen, Germany), internal tank dimensions (length, width, height; mm) 500 × 290 × 130, filled with 14 L deionized water, and regulated at four different temperatures (55, 65, 75 and 85 °C). These temperatures were selected for calibration checking of the digital food thermometer because they cover the range of coffee serving temperatures found in the German food service industry [22].

Given possible inaccuracy in the water bath temperatures, we used, as a standard thermometer, a calibrated mercury filling glass thermometer (−10 °C to +110 °C range; 1 °C division; 303 mm length; Alla France, Chemillé, France) in parallel with the digital thermometer. The calibration of the glass thermometer was checked and confirmed in advance through 5 measurements of boiling ethanol (99.9% purity, boiling point 78 °C, J. T. Baker, Mexico) and chloroform (99.8% purity, boiling point 61 °C, Alphatec, Brazil).

Water bath temperatures from the glass thermometer (calibration standard) and the digital food thermometer were collected in 10 intraday replicate measurements at the same time and proximity in the thermostatic bath, and the coefficient of variation (CV) never exceeded 0.3%. By plotting the data (glass thermometer, x-axis; digital thermometer, y-axis) in a Microsoft Excel 2013 XY scatter graph, a mathematical relationship was obtained and expressed by the linear equation y = 1.037x − 2.151, with R^2^ = 0.9998. The coefficient of determination, or R^2^, indicates how well the trend line corresponds to the data (the closer the R^2^ to 1, the better the fit). Although the results from the digital thermometer almost perfectly matched the standard glass thermometer, we decided to use the equation to make minor corrections in the temperatures from the digital thermometer (i.e., for a given “y” value, a corrected “x” value was obtained from the equation). Thus, the data from the digital thermometer stated in the results section are already corrected by the equation.

### 2.4. Temperature Measurements of Best-Selling Coffee-Based Hot Beverages in the Food Service Industry

On arrival at the coffee shop (CS) or low-cost food service establishment (LCFS), the research staff, acting as customers, either sat at a table (CS) or stood at the counter (LCFS), asked the attendant what the most frequently ordered coffee-based beverage (type and volume) was. The “bestselling” beverage was then ordered. The thermometer’s sensor was immersed in the beverage as soon as it was handed over. Once the temperature in the thermometer stabilized (i.e., reached a plateau and started to fall; this took approximately 1 min), the upper value was recorded (one measure per coffee per establishment). No sugar, sweetener, milk or water was added to the beverage and therefore it represents the usual serving temperature, or immediate consumption temperature, in a “worst-case scenario”. Besides coffee type/volume and temperature, information such as cup material (porcelain, glass, paper, plastic, etc.), beverage price, addition of sugar in the preparation of the beverage (sweetened or unsweetened), type of storage (in cases where the bestselling beverage had been brewed previously and stored ready-to-serve, a procedure typically used in LCFS), type of coffee machine (portafilter machine, capsule machine, etc., typically used in CS for coffee brewing) and ambient temperature were also recorded. Regarding specific construction materials of disposable plastic cups, the information was collected from plastic abbreviations (polypropylene, PP; polystyrene, PS, etc.) which is stamped at the bottom of every cup. The work’s main objective (“to investigate the usual serving temperatures of coffee in the food service industry”) was given to the attendant or manager afterwards. All ordered coffee beverages were paid for by the research staff.

### 2.5. Cooling Profile of Hot Water Systems to the IARC Threshold Temperature and Beyond

According to the Brewing Control Chart of the Specialty Coffee Association [29], the solubles concentration in brewed coffee ranges from 0.8% to 1.60%, with a median concentration of 1.20% (1.2 g in 100 mL), which corresponds to the brew strength that should be present in the so called “golden cup standard“ [30]. By assuming 1.2% soluble concentration as the reference, a cup of brewed coffee is about 99% water, therefore pure water is a reasonable model to simulate brewed coffee in cooling behavior studies. Moreover, recent experimental results from Langer et al. [31] show that coffee has a similar cooling behavior as hot water.

Cooling behavior studies (temperature as a function of time) to 65 °C, and beyond, were carried out using hot deionized water under different conditions (water temperature, cup construction material and ambient temperature) to simulate hot coffee beverages found in the field work. A water volume of 50 mL was used for all cooling studies. As the working cup volume and materials, a 50 mL disposable white polystyrene (PS) cup (SM-050 model, Copobras S/A, João Pessoa, PB, Brazil; top external diameter = 52 mm; base external diameter = 34 mm; height = 41 mm; wall thickness ~0.3 mm) and a 50 mL white porcelain cup (prisma model, Schmidt Porcelain, Pomerode, SC, Brazil; top external diameter = 58 mm; base external diameter = 34 mm; height = 57 mm; wall thickness ~4.9 mm) were chosen. Initial water temperatures in PS cups of 77 °C and 70 °C, both under an ambient temperature of 30 °C, were chosen to represent higher and median coffee temperatures found in the low-cost food service establishments. Initial water temperatures in porcelain cups of 75 °C and 69 °C, both under ambient temperature of 25 °C, were chosen to simulate higher and median coffee temperatures found in the coffee shops.

We managed to achieve the mentioned initial water temperatures by placing the cup filled with 50 mL water in the thermostatic bath regulated at the specific temperatures. As the water tank depth was higher than the cup’s height, we managed to keep it surrounded by water and at the same time stabilized by raising the tank bed with a support. Once the cup was stable, the sensor of the digital thermometer sensor was fixed in it. As the desired temperature in the cup was reached (77, 75, 70 or 69 °C), a chronometer was initiated and, right after, the cup was removed from the tank, wiped with a paper towel (external surface) and placed on the lab bench. Next, the sensor was fixed again in the cup and the cooling temperatures collected in one minute intervals up to 10 min. The procedure for each model system (initial temperature, cup type and ambient temperature) was carried out in three replicates and the CV never exceeded 1.9%.

## 3. Results

### 3.1. Coffee Temperatures in Low-Cost Food Service Establishments and Coffee Shops

Table 1 presents an overview of the bestsellers’ serving temperatures from 50 low-cost food service establishments (LCFS). Detailed results from each establishment, such as coffee volume, price, type of storage (thermos flask or electric dispenser) and cup material, are presented in Table A1 (Appendix B).

According to Table 1, the temperature range and median value of all best-selling coffees were 62–78 °C and 70 °C, respectively. A total of 90% of the collected values exceeded the IARC threshold of 65 °C. The 95th percentile was 77 °C (95% of temperature data are below this value) and it represents a reasonable estimate of a particular high temperature found in the LCFS. The bestsellers in the LCFS were dominated by sweetened black coffee (68% of total) followed by unsweetened black (26%) and sweetened milk coffee (6%).

It is of interest to highlight that all ordered best-selling coffees in the LCFS had been brewed in advance and stored ready-to-serve in either a thermos flask (82% of samples) or an electric hot coffee dispenser (28%). The mean ± standard deviation of all serving coffee volumes and prices (in Brazilian reais, BRL) was 86 ± 44 mL (median = 50 mL) and BRL 0.84 ± 0.45 (median = BRL 0.50), respectively. Regarding the serving cup material, 66% were made of disposable plastic (40%, polystyrene; 26%, polypropylene), 26% of glass and 8% of porcelain (Table A1).

With respect to the most common coffee in LCFS—sweetened black coffee (34 samples)—62% were ordered in 50 mL shots and 38% in 125–150 mL. The majority of the 50 mL shots of sweetened black coffee were served in 50 mL (approximate working volume) polystyrene (PS) cups (48%), followed by polypropylene (PP) of the same working volume (38%). Thus, a 50 mL shot of sweetened black coffee served at 70 °C in 50 mL PS cups is probably the best representative of the bestseller coffee in LCFS establishments (Table A1).

An overview of the bestsellers’ serving temperatures from 50 coffee shops (CS) is shown in Table 2. Detailed results from each CS establishment, such as coffee volume, price, type of machine (used for coffee brewing) and cup material, are presented in Table A2 (Appendix B).

According to Table 2, the temperature range of all best-selling coffees in CS was 52–85 °C. Median (69 °C) and 95th percentile (75 °C) were close to the values found in LCFS. However, in comparison to LCFS, a smaller percentage (68%) of the collected temperatures exceeded the IARC threshold of 65 °C. With respect to cappuccino, particularly high maximum values (85 °C) and P95 (79 °C) were observed; it is noteworthy that hot beverages spills that occur over approximately 82 °C are clearly hazardous and likely to lead to mid-dermal burns which can be serious [32]. The bestsellers in the CS were dominated by espresso (66% of total) followed by cappuccino (28%) and other coffee-based beverages (6%) (Table 2).

All best-selling coffees in CS were freshly brewed on ordering and portafilter machines (espresso machine) were the dominant type (90%) in the establishments. The mean ± standard deviation of all serving coffee volumes and prices (in Brazilian reais, BRL) was 85 ± 59 mL (median = 50 mL) and BRL 5.41 ± 1.95 (median = BRL 4.50), respectively. Regarding the serving vessel material, 88% were made of porcelain and 12% of glass (Table A2).

With respect to the most common coffee in CS (espressos, 33 samples), most of them (97%) were ordered in 50 mL shots while the sizes of cappuccinos (14 samples) were more diverse (150–160 mL, 50%; 225–250 mL, 29%; 50 mL, 21%). A total of 94% of the espressos were extracted in portafilter machines. Regarding cup material, 88% of ordered coffees were served in porcelain, 12% in glass and all 50 mL shots of espresso were served in 50 mL (approximate working volume) porcelain cups. A 50 mL shot of espresso served at 69 °C in 50 mL porcelain cups is therefore the best representative of the bestseller coffee in CS establishments (Table A2).

### 3.2. Cooling Behavior of Hot Water Systems to the IARC Threshold Temperature and Beyond

Given that drinking beverages with temperatures above 65 °C is a habit classified by the IARC as “probably carcinogenic to humans”, this section aimed to obtain practical results for coffee consumers regarding safe drinking. To achieve that, we studied the cooling behavior of hot water system representatives of coffees served at LCFS and CS, and these included variations in cup materials (disposable PS plastic and porcelain), ambient temperatures (30 °C and 25 °C) and initial temperatures (medians and P95). As the cooling process of a hot liquid poured into a drinking vessel is influenced by factors such as the thermal conductivity of the cup material, wall thickness of the cup, liquid temperature, cooling surface areas (of both the cup and open surface of the liquid) and ambient temperature, differences were expected to arise in the results.

Figure 1 shows the cooling behaviors of two hot water system representatives of coffees served at the LCFS. The two LCFS scenarios consisted of 50 mL water in PS cups at initial temperatures of 77 °C (P95) and 70 °C (median), both under a controlled ambient temperature of 30 °C.

Figure 2 presents the cooling behaviors of two hot water system representatives of coffees served at the CS. They consisted of 50 mL water in porcelain cups at initial temperatures of 75 °C (P95) and 69 °C (median), both under a controlled ambient temperature of 25 °C.

Despite the variations in factors affecting heat transfer in our systems, a relatively small difference was observed when the systems were studied at the P95 zone (77 °C/PS and 75 °C/porcelain). In this higher zone of temperature, the cooling time to the safe threshold (65 °C) was 3:12 min (Figure 1) and 2:48 min (Figure 2) in the PS and porcelain cup systems, respectively. Furthermore, when the initial temperatures of the systems were at the median zone (70 °C/PS and 69 °C/porcelain), the cooling behavior of the PS and porcelain cup systems were similar: both systems took 1:36 min to 65 °C (Figure 1 and Figure 2), which may be rounded up to 2 min.

Although the results suggest similar to close cooling behavior from the porcelain and PS systems, the thermal conductivities of porcelain and PS are 1.5 W m^−1^ K^−1^ [33] and 0.12 W m^−1^ K^−1^ [34], respectively, thus PS insulates about 12 times better than porcelain. On the other hand, the wall thickness of the porcelain and PS vessels in our study are 4.9 and 0.3 mm, respectively, therefore the porcelain cup would have a 16-fold insulation capability advantage in this respect. To complicate comparisons further, the ambient temperatures under which the systems were studied and the shapes of the cups (hence their cooling surface areas to the outer surface) are different. However, in an attempt to clarify the differences in insulation capabilities of the PS and porcelain cups used in our study, we decided to carry out an experiment in which all factors affecting heat transfer remained constant (initial temperature and ambient temperature) except the cups themselves. Figure 3 shows the cooling behaviors of two hot water systems from 77 °C in 50 mL porcelain and polystyrene (disposable) cups, both at an ambient temperature of 25 °C.

Under our experimental conditions and once the temperatures of the water and the cups are equalized, Figure 3 shows that the porcelain cup is a better insulator than the PS cup. The cooling time from 77 °C to 65 °C was about 3:12 min and 4:00 min in the PS and porcelain cup systems, respectively (48 s difference). Interestingly, the differences in insulating capabilities of the cups over time (up to 10 min) were more noticeable; for instance, the cooling time to 60 °C was approximately 4:48 and 6:00 min in the PS and porcelain cup systems, respectively (1:12 min difference). In brief, the results in Figure 3 suggest that the larger wall thickness of the porcelain cup probably played a major role in compensating the higher thermal conductivity of porcelain (compared to the PS material). However, for a proper and detailed comparison of the insulation capabilities of the cups, their shape would have to be the same, but this is beyond the scope of the present study.

## 4. Discussion

This study recorded the serving temperatures of the best-selling coffee-based beverages in 100 food service establishments (50 low-cost food service and 50 coffee shops) in Brazil. The bestsellers in the low-cost food service establishments (LCFS) were dominated by 50 mL shots of sweetened black coffee, which were stored ready-to-drink in thermos flasks and typically served in 50 mL disposable polystyrene cups. In the coffee shops (CS), a 50 mL shot of freshly extracted espresso from portafilter machines, served in 50 mL porcelain cups, was the dominant beverage. The results show that the majority of best-selling coffee beverages are served very hot (>65 °C) in both segments of the food service industry, but a much larger proportion was observed in the LCFS (90%) than in the CS (68%).

This is the first published study on the serving temperature of coffee beverages in the Brazilian food service industry, thus comparisons to previous local studies are not possible at present. In the last 20 years, two research papers on the serving temperature of coffee beverages in food service establishments have been published, one carried out in Germany [22] and the other in the USA [21], but the German establishments seem to be the only ones that bear resemblances to the CS segment studied here (the establishments assessed in the USA were quick service restaurants, such as McDonald’s, Burger King and KFC). However, the other segment of the Brazilian food service industry—the LCFS—is probably unique and therefore incomparable to the German and American establishments.

The mean value (± standard deviation) of the measured temperatures in the Brazilian CS (67 ± 7 °C; 50 samples) was below the value found in the German food service industry (75 °C ± 5 °C; 356 samples) and, as a result, the latter achieved a higher percentage of samples above 65°C (98%) [22]. Although the German study measured a much larger number of samples, the types of coffees were not disclosed, therefore specific comparisons cannot be made. However, since 83% of the machines used for coffee brewing in the German food service industry were either portafilter (37%) or fully automated machines (46%), the measured samples were probably Americano, Café crème, espressos or milk-based espresso drinks, such as cappuccinos and lattes. The coffee samples obtained specifically from portafilter machines showed an even higher mean value (77°C ± 5 °C) in the German study; a proposed explanation was the fact that the portafilter machines exhibited pressure tank settings of extremely high temperatures (120 °C), information which is not available in the CS.

Although the majority of the coffee samples in the CS were above the IARC threshold of 65 °C, it can be said that, with differences between the German study and this research kept apart, the Brazilian CS scenario of coffee temperatures is less worrying for consumers than the situation found in the German food service industry. Moreover, most coffee (68%) drunk in the CS were espressos (arithmetic mean, 68 °C; median, 69 °C), therefore it is likely that by simply adding sugar to the beverage (a very common habit in the Brazilian coffee culture), the temperature will drop from 69 °C to below 65 °C. However, in a worst-case scenario (i.e., no addition of sugar and no waiting time once the drink is handed over), the serving temperatures in the CS still pose a hazard to Brazilian consumers.

Regarding the hazard, our findings on the cooling time of water systems representing a typical coffee in the CS (50 mL shot of espresso served at 69 °C in 50 mL porcelain cups and at an ambient temperature of 25 °C) show that consumers should allow a minimum cooling time to 65 °C of 1:36 min, as seen in Figure 2. However, by taking into consideration the recent suggestion showing that consumers perceive coffee most preferable at temperatures of about 63 °C [28], compounded by the fact that the coffee roast beans used in the CS are probably of high sensory quality (“gourmet” or even “specialty” coffee beans), consumers of espresso in CS would enjoy the beverage to the maximum by waiting 2:12 min instead of 1:36. All things considered, including a certain level of uncertainty, a minimum cooling time before drinking of approximately 2 min is recommended.

In comparison to the CS, the distinction of the coffee commonly served in the LCFS (sweetened black coffee stored in thermos flasks) seems to present an increased hazard to Brazilian consumers, as discussed next.

It has been reported by a cross-sectional population-based survey that the most common coffee preparation method in Brazilian households is filter brewing (86%), with a reusable cloth strainer as the dominant type of filter (63%) followed by disposable paper (23%) [35]. The survey also reported that in many circumstances, coffee is prepared in family-size quantities and stored with sugar in thermos flasks in households, a situation we also found in the LCFS establishments.

Although the preparation methods behind the coffees were not specifically targeted in this research, it is reasonable to assume that the sweetened black coffee offered in the LCFS follows the brewing procedure typically carried out in many Brazilian households (i.e., cloth filter brewing, sugar addition and storage in thermos), but details on the preparation were not quite clear. However, a search in a video-sharing platform for “traditional Brazilian cloth-filter brewing coffee recipes”, associated with the addition of sugar and storage in thermos flasks, offered some details of the procedure. In brief, a standard recipe for such coffee typically consists of the following: (1) add ~750 mL water into a pot; (2) heat the pot on a stove until the contents simmer; (3) add ~35 g of ordinary ground coffee (fine to medium-fine grind size) and ~120 g granulated sugar to the simmering water and stir; (4) bring the brew to boiling or near boiling (the mixture will froth up at this point); (5) remove the pot from the heat and (6) pour the sweetened hot coffee brew through a reusable cloth strainer (with the handle) directly into a thermos flask of adequate volume. After straining, the yield is about 700 mL of cloth-filtered sweetened black coffee. Instead of adding the sugar together with the coffee into the pot, it may be added between steps 5 and 6. Moreover, a paper filter in a holder may be used instead of a cloth strainer. According to the recipes, the concentration of sugar is about 15 g in 100 mL and the coffee-to-water ratio is 1 to 21 (4.7 g ground coffee/100 mL water). It is of interest to mention that in some LCFS establishments, there were the alternatives of sweetened milk coffee and unsweetened black coffee ready-to-serve in thermos flasks (in parallel to sweetened black coffee), thus we understand that these beverages probably follow the brewing procedure of the sweetened black coffee with minor changes in the recipe.

Due to the nature of the coffee normally used in the traditional recipe of sweetened black coffee (lower price; lower sensory grades of certifications or even non-certified coffee; dark to very dark roasts; etc.), compounded by the brewing procedure itself, the resulting beverage would probably taste quite bitter had no generous quantities of sugar been included in the recipe. Thus, the likely role of sugar in the recipe besides bringing sweetness is to balance the harshness of the brew.

The procedure likely adopted in LCFS for sweetened black coffee brewing poses two main problems for consumers. The first is that the thermos is likely filled with filtered sweetened black coffee in scalding temperatures, thus the first consumers may experience an extremely hot beverage, particularly if the thermos is preheated with boiling water. The second is that the beverage is prepared to be ready to serve and ready to drink, therefore once poured into a cup, there will be no impacting cooling aid into the system (for instance, the addition of sugar) other than the heat loss from the cup system over time. Cold milk was not mentioned as a cooling aid because in Brazil, contrary to many European countries, if milk is to be added to a cup of black coffee, it is often heated beforehand.

Regarding the heat loss, when coffee is poured into a cup, it undergoes a complex transient cooling process, which is determined by the following superimposed sub-processes: heat transfer to the inner surface of the cup, heat conduction and heating and subsequent cooling of the cup material, heat transfer to the outer surface of the cup by convection and radiation and heat transfer to the open surface of the coffee by convection, radiation and evaporation [22]. In turn, the heat transfer processes are influenced by factors such as the type of cup material, its width, the heat transfer area of the cup and the ambient temperature.

Our findings on the cooling time of water systems representing a typical coffee in the LCFS (50 mL shot of sweetened black coffee at 70 °C in 50 mL PS cups) show that consumers should allow a minimum cooling time to 65 °C of approximately 2 min, as seen in Figure 1.

Although the cooling studies of water systems at 77 °C has showed that a standard 50 mL disposable PS cup cools down more easily than a standard 50 mL porcelain cup, it should also be considered that in a real coffee system, the presence of generous quantities of sugar in the beverage (such as the case of sweetened black coffee served in the LCFS) may have an impact on heat loss. This is because the thermal conductivity of an aqueous solution decreases as the concentration of sucrose increases. For instance, the thermal conductivity of pure water at 60 °C is 0.654 W m^−1^ K^−1^ [33] (p. 694) while that of a 20% (m/m) sucrose solution at same temperature is 0.581 W m^−1^ K^−1^ [36] (p. 560). It would be interesting to carry out further cooling investigations on the effect of sugar (already present in a solution or by adding it afterwards) in hot water systems.

Apart from additional investigations of the effect of sugar on cooling, a number of future researches may be carried out from this point. First, and most importantly, data collection on coffee temperature should be expanded to Brazilian households and workplaces, but a second round of data from the food service industry, preferably in a different Brazilian location, would also be interesting to confirm our initial findings. In addition, research should be done on the drinking behaviors of Brazilian coffee consumers (how many cups, how fast it is drunk, what is the frequency, what is added to the coffee and how much, where it is drunk, among others). Finally, wherever possible, future research should also investigate the coffee method preparations on all consumption segments (food service, households and workplace), so that a full picture of coffee consumption can be drawn with corresponding intervention measures proposed.

A more complete picture of coffee consumption habits and the temperature at which consumption commonly occurs in Brazil would also present an excellent opportunity for an esophageal cancer risk assessment of hot beverages consumption in the country, with special consideration to coffee.

## Figures and Tables

**Figure 1 foods-09-01047-f001:**
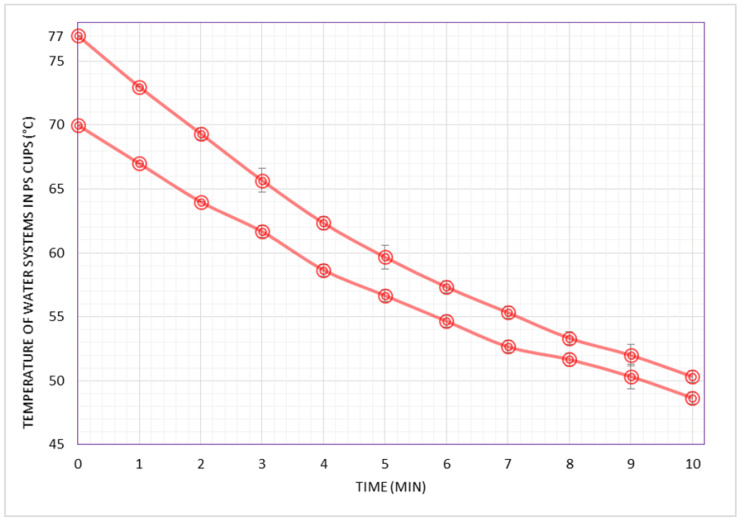
Cooling behaviors (temperature as a function of time) of hot water systems from 77 °C and 70 °C, both in disposable 50 mL polystyrene (PS) cups and at a constant ambient temperature of 30 °C. Temperatures of the water systems were collected in one minute intervals up to 10 min. Point values are shown as the mean of 3 repetitions ± standard deviations of each experiment and the coefficient of variation never exceeded 1.9%.

**Figure 2 foods-09-01047-f002:**
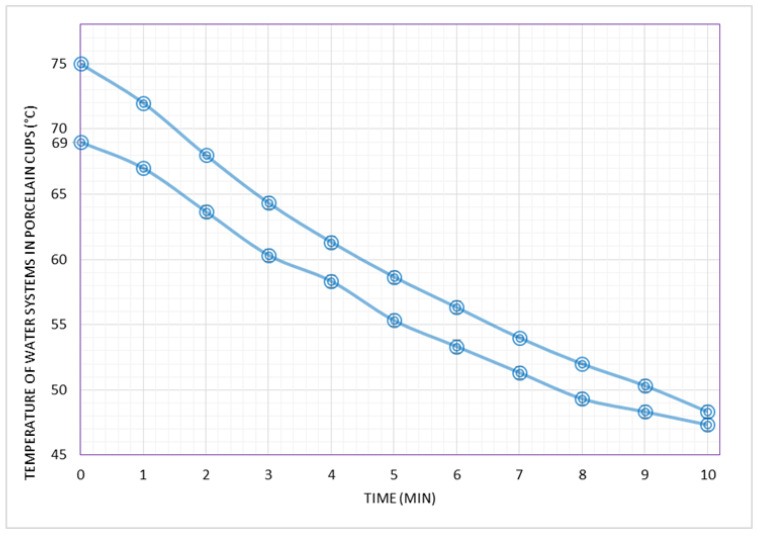
Cooling behaviors (temperature as a function of time) of hot water systems from 75 °C and 69 °C, both in 50 mL porcelain cups and at a constant ambient temperature of 25 °C. Temperatures of the water systems were collected in one minute intervals up to 10 min. Point values are shown as mean of 3 repetitions ± standard deviations of each experiment and the coefficient of variation never exceeded 1.0%.

**Figure 3 foods-09-01047-f003:**
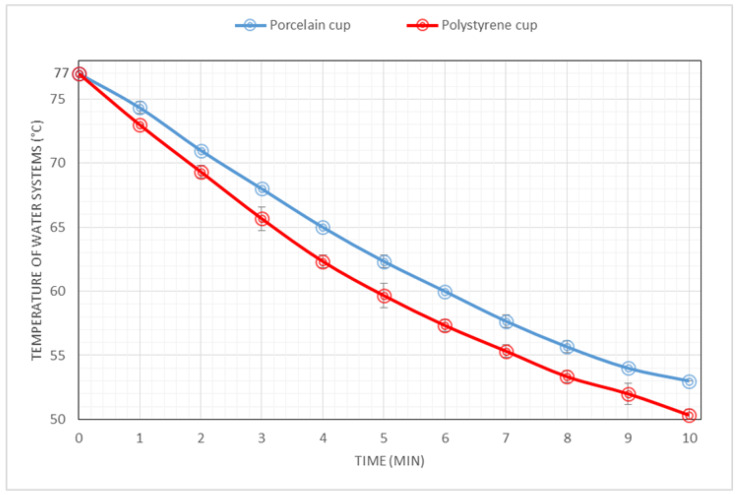
Cooling behavior (temperature as a function of time) of hot water systems from 77 °C in 50 mL porcelain and polystyrene (disposable) cups, both at ambient temperature of 25 °C. Temperatures of the water systems were collected in one minute intervals up to 10 min. Point values are shown as mean of 3 repetitions ± standard deviations of each experiment and the coefficient of variation never exceeded 1.6%.

**Table 1 foods-09-01047-t001:** Summarized results of bestsellers’ serving temperatures (°C) in 50 low-cost food service establishments (LCFS), according to coffee type ^1^.

Coffee Type ^2^	Min. ^3^	Max. ^4^	Mean	Median ^6^	P95 ^7^	%
			**± SD ^5^**			**>65°C ^8^**
Total	62	78	70 ± 4	70	77	90
*n* = 50						
Sweetened black	62	78	69 ± 4	70	76	88
*n* = 34						
Unsweetened black	66	77	71 ± 3	71	77	100
*n* = 13						
Sweetened milk coffee	64	74	69 ± 4	69	74	67
*n* = 3						

^1^ For additional information on data collected from each LCFS (bestseller coffee, volume, price, cup material, etc.), see Table A1 in Appendix B; average ambient temperature in LCFS was 31 °C and all establishments were non-climatized (not equipped with a controlled air temperature). ^2^ The ordered beverage was always the establishment’s bestseller, according to information provided by the attendant previously. ^3^ Minimum temperature (°C). ^4^ Maximum temperature (°C). ^5^ Mean temperature ± standard deviation (°C). ^6^ Median temperature (°C). ^7^ 95th percentile (°C). ^8^ Percentage of samples with serving temperatures above 65°C.

**Table 2 foods-09-01047-t002:** Summarized results of bestsellers’ serving temperatures (°C) in 50 coffee shops (CS), according to coffee type ^1^.

Coffee Type ^2^	Min. ^3^	Max. ^4^	Mean	Median ^6^	P95 ^7^	%
			**± SD ^5^**			**>65°C ^8^**
Total	52	85	67 ± 7	69	75	68
*n* = 50						
Espresso	54	77	68 ± 6	69	75	76
*n* = 33						
Cappuccino	52	85	68 ± 9	71	79	57
*n* = 14						
Other coffee drinks	57	65	60 ± 3	59	64	33
*n* = 3						

^1^ For additional information on data collected from each CS (bestseller coffee, volume, price, cup material, etc.), see Table A2 in Appendix B; average ambient temperature in CS was 25 °C and all establishments were fully climatized. ^2^ The ordered beverage was always the bestseller, according to information provided by the attendant previously. ^3^ Minimum temperature (°C). ^4^ Maximum temperature (°C). ^5^ Mean temperature ± standard deviation (°C). ^6^ Median temperature (°C). ^7^ 95th percentile (°C). ^8^ Percentage of coffee samples with serving temperatures above 65°C.

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
