# Peer review of "Serving Temperatures of Best-Selling Coffees in Two Segments of the Brazilian Food Service Industry Are “Very Hot”"

_foods, 2020, doi:10.3390/foods9081047_

Round 1
Reviewer 1 Report
This is an engaging article that purposefully question our knowledge of the subject. The metodology is robust and I observed a good readability of results and discussion.
The manuscript specifically investigated the serving temperatures of coffee brews in Brazil, due to the health-related problem of hot beverages, being possibly carcinogenic for humans. The main theme of the study is really interesting and innovative because, although we have at disposal a wide literature on consumer preferences on serving temperatures and the effect of temperature on some physico-chemical and sensorial properties of coffee brews, the question of temperature related to health effects is understudied. Actually, except for the mentioned German study, no data on serving temperature in Brazil as well as in other parts of the world are available. So, the methods used in this paper, which are clear and reproducible, could be adopted by other research groups to investigate serving temperatures of other types of coffee brews (e.g. Americano, Turkish, etc.) or in other countries. The cooling behaviour investigated in this paper is obviously a model system, but it gives useful information for future investigation which could be implemented by using coffee beverages as well.
The manuscript is well written, concise, easy to read. Results are adequately described and focused on the most important and relevant data. Discussion is fluent and appropriate, conclusions are well supported by results and satisfactorily match the main objective of the study.
Author Response
R1 is thanked for the encouraging remarks, in particular because this is the first time our group in Brazil is working on the subject. R1 is also thanked for providing an excellent and adequate description of what we did in our study.
Reviewer 2 Report
General Comments: The paper is interesting and raises and important issue, coffee temperature, from an important perspective, viz., hot beverage consumption and cancer. The authors have, however, ignored a large body of literature that focuses on customer preferred beverage temperatures and recent studies focused on hot beverage consumption and cancer.
Please consider including a discussion of:
https://doi.org/10.1002/ijc.32220
https://doi.org/10.7326/M17-2000
https://doi.org/10.1016/j.foodres.2018.08.014
https://www.acpjournals.org/doi/10.7326/M17-2000#s2-M172000
https://doi.org/10.1375/jhtm.14.1.37
https://doi.org/10.1016/S0950-3293(98)00053-6
https://www.mskcc.org/news/burning-issue-truth-about-hot-drinks-and-esophageal-cancer-risk
https://doi.org/10.1111/j.1365-2621.2002.tb08814.x
Specific Comments:
page, line
2, 47-48: Does your estimate and discussion of coffee volume take into consideration the varying normative/typical strength of the brew across cultures/countries?
2, 71-77: Please explain MOE for the uninitiated reader.
3, 145: What is the basis for this decision?
4, 169ff: Does water and coffee lose temperature at the same rate?
7, Figure: Please provide a better figure/legend. It is hard to read/follow the figure. (Ditto page 8)
7, 287ff: Please elaborate more on the notion of insulation capability and thickness of cup. How does the shape of the cup impact insulation/temperature loss?
10, 381: Basis for "might consider?"
Author Response
General Comments: The paper is interesting and raises and important issue, coffee temperature, from an important perspective, viz., hot beverage consumption and cancer. The authors have, however, ignored a large body of literature that focuses on customer preferred beverage temperatures and recent studies focused on hot beverage consumption and cancer.
R2 is thanked for the contribution to our paper in the form of comments, questions and the suggested references. As the text was written with a certain logical sequence in mind, particularly the introduction, we have attempted to accommodate some of the suggested references without causing too much disruption to the flowing of the original text. All revisions/changes/corrections in the text are highlighted in red.
Please consider including a discussion of:
https://doi.org/10.1002/ijc.32220
https://doi.org/10.7326/M17-2000
The above references (Islami et al., 2020; Yu et al., 2018) were included in the context of the 4th paragraph (Introduction, lines 63-78)
https://doi.org/10.1016/j.foodres.2018.08.014
This reference was not included as being not completely relevant.
https://www.acpjournals.org/doi/10.7326/M17-2000#s2-M172000
This is the same reference as the 2nd one.
https://doi.org/10.1375/jhtm.14.1.37
https://doi.org/10.1016/S0950-3293(98)00053-6
These two references (Borchgrevink et al. 1999, 2007) were included as requested.
https://www.mskcc.org/news/burning-issue-truth-about-hot-drinks-and-esophageal-cancer-risk
This is a press article, the underlying reference is 1st ref above, which was included.
https://doi.org/10.1111/j.1365-2621.2002.tb08814.x
This reference was included.
In total, we have included the following new references in the main text:
Borchgrevink, C.P.; Sciarini, M.P.; Susskind, A.M. Hot beverages at quick service restaurant (QSR) drive-thru windows. J. Hosp. Manage. Tourism 2007, 14, 37-46, doi:10.1375/jhtm.14.1.37.
Abraham, J.; Diller, K. A review of hot beverage temperatures-satisfying consumer preference and safety. J. Food Sci. 2019, 84, 2011–2014, doi: 10.1111/1750-3841.14699.
Borchgrevink, C.P.; Susskind, A.M.; Tarras, J.M. Consumer preferred hot beverage temperatures. Food Qual. Prefer. 1999, 10, 117–121, doi:10.1016/S0950-3293(98)00053-6.
Pipatsattayanuwong, S.; Lee, H.S., Lau, S.; O’Mahony, M. Hedonic R-index measurement of temperature preferences for drinking black coffee. J. Sens. Stud. 2001, 16, 517–536, doi:10.1111/j.1745-459X.2001.tb00317.x.
Lee, H.S.; O’Mahony, M. At what temperatures do consumers like to drink coffee? Mixing methods. J. Food Sci. 2002, 67, 2774–2777, doi:10.1111/j.1365-2621.2002.tb08814.x.
Brown, F.; Diller, K.R. Calculating the optimum temperature for serving hot beverages. Burns 2008, 34, 648–654, doi:10.1016/j.burns.2007.09.012.
Specialty Coffee Association. Brewing Control Chart. Available online: https://store.sca.coffee/products/brewing-control-chart-grams-per-liter?variant=14732978758 (Accessed on 20 July 2020).
Specialty Coffee Association. Brewing standards. Available online: https://sca.coffee/research/coffee-standards (Accessed on 19 July 2020).
Langer, T.; Winkler, G.; Lachenmeier, D.W. Untersuchungen zum Abkühlverhalten von Heißgetränken vor dem Hintergrund des temperaturbedingten Krebsrisikos. Deut. Lebensm. Rundsch. 2018, 114, 307–314. (In German), doi:10.5281/zenodo.1402982.
Abraham, J.P.; Nelson-Cheeseman, B.B.; Sparrow, E.M.; Wentz, J.E.; Gorman, J.M.; Wolf, S.E. Comprehensive method to predict and quantify scald burns from beverage spills. Int. J. Hyperther. 2016, 32, 900–910, doi:10.1080/02656736.2016.1211752.
Specific Comments:
page, line
2, 47-48: Does your estimate and discussion of coffee volume take into consideration the varying normative/typical strength of the brew across cultures/countries?
R2 is thanked for the question. In the 2nd paragraph (Introduction, lines 44-45, we stated that a coffee-to-water ratio of 1 to 12.5 (8 g roasted coffee/100 ml water), obtained from references 10 (p.25) and reference 11 (p. 3), was applied to convert roasted ground coffee into liquid coffee.
In reference 10 (Farah, 2012), page 25, it is stated the following: “The proportion of coffee to water varies considerably in different countries and according to individual preferences, but is usually 8–20 g coffee/100 mL water.“
On the other hand, reference 11, page 3, states the following: “For example, while in most European countries, in the USA and Canada the use of ~7 g (1/4 oz) per 100 mL is common for filtered coffees, in Brazil 10 g or more are used.“
As seen, we applied the lower limit, 8 g/100 ml, from the world range (8-20g/100 ml) presented in reference 10. We chose specifically 8 g/100 ml because it is also close to the proportion used for filtered coffees (a common brewing procedure) in most European countries, in the USA/Canada (~7g/100 ml) and Brazil (10 g/100 ml), according to reference 11. Ideally, we should have searched the typical coffee-to-water ratios used in the countries and applied them accordingly. However, this search would take an appreciable time and is beyond the scope of our study. Thus, aiming at obtaining a reasonable volume of coffee that is drunken worldwide, 8 g/100 ml, which is equivalent to a coffee-to-water ratio of 1 to 12.5, seems to be an acceptable proportion.
2, 71-77: Please explain MOE for the uninitiated reader.
The reviewer 2 is right, MOE (Margin of Exposure) is not an approach known by many readers and this is why we attempted to explain it, but it was probably too concise. We have included additional sentences in the 5th paragraph (Introduction, lines 79-90) that hopefully provides a clear and fuller explanation of the MOE.
3, 145: What is the basis for this decision?
The reason is that a 1 °C difference was observed at the highest assessed temperature, 83 °C (i.e. when the standard glass thermometer recorded 83 °C in the water bath, the digital thermometer recorded 84 °C). The other three assessed temperatures in the standard glass thermometer (54, 64 and 74) matched the values in the digital thermometer. By using the linear equation we were able the correct the digital thermometer at more extreme temperatures eventually found in the food service industry. It should be noted that some of the recorded temperature were in the region of 80 °C (the maximum value was 85 °C), therefore we understand the correction was necessary.
4, 169ff: Does water and coffee lose temperature at the same rate?
Coffee and water have very similar cooling behavior. We have included the following text in the beginning of section 2.5 (Materials and Methods, lines 190-196) to support the use of water as a model for coffee.
“ According to the Brewing Control Chart of the Specialty Coffee Association [29], solubles concentration in brewed coffee ranges from 0.8% to 1.60%, with a median concentration of 1.20% (1.2 g in 100 ml), which corresponds to the brew strength that should be present in the so called “golden cup standard“ [30]. By assuming 1.2% soluble concentration as reference, a cup of brewed coffee is about 99% water, therefore pure water is a reasonable model to simulate brewed coffee in cooling behavior studies. Moreover, recent results from Langer et al. [31] show that coffee has a similar cooling behavior as hot water. “
7, Figure: Please provide a better figure/legend. It is hard to read/follow the figure. (Ditto page 8)
Given the difficulties experienced by R2, we have decided to split Figure 1 (1st submission) in two separate figures: one for the PS systems (LCFS), Figure 1, and other for the porcelain systems (CS), Figure 2. Due to this change, the Figure 2 in the 1st submission is now presented as Figure 3. All revisions in section 3.2 (cooling behavior of hot water systems to the IARC threshold temperature and beyond), lines 288-352, are highlighted in red.
7, 287ff: Please elaborate more on the notion of insulation capability and thickness of cup. How does the shape of the cup impact insulation/temperature loss?
The reviewer raises an interesting point. We added extra sentences in lines 325-352 to address this point. We have decided that the approach would be concise because the paper is not focused on heat transfer/engineering. Please note that in lines 441-447 (Discussion) several aspects of the heat loss (in the context of a cup of hot coffee) are briefly presented as well.
10, 381: Basis for "might consider?"
Taking into account the LCFS establishments adopt simple preparations/brewing procedures, we understand that the typical Brazilian brewing method described for the sweetened black coffee (filter brewing with reusable cloth strainer, addition of sugar, storage in thermos) is probably used for the other two types of coffees stored in thermos flasks (sweetened milk coffee and unsweetened black coffee). Moreover, in principle it is easy to adapt the preparation of sweetened milk coffee and unsweetened black coffee to the sweetened black coffee recipe. For instance, for the sweetened milk coffee it is just a matter of replacing water – total or part of it - by milk in the recipe; for the unsweetened black coffee, it is just a matter of not adding sugar in the procedure. For clarity, however, we have changed the sentence slightly in lines 425-426.
Round 2
Reviewer 2 Report
You have improved the manuscript but did not integrate the literature on preferred hot beverage temperatures into your manuscript. It appears as if the new literature was added for the reviewer rather than the potential reader. It would be interesting if you discussed the apparent overlap of preferred consumption temperatures with temperatures causing harm. Rather than dismiss other data based on the type of establishments from which the coffee was attained, focus on your main issue, temperature and cancer!
Author Response
Thank you for your additional comments on our paper. Actually, the authors believe that the additional request of R2 is a bit too demanding. It must be considered that we 100% fulfilled all requests during the first revision, and it must additionally be considered that this is not an epidemiological study, so we did not measure cancer as outcome, apart from studying the exceedance of the threshold temperature suggested by the IARC monographs working group in Vol. 116. The authors understand, however, that our results, together with others, may serve as basis for a cancer risk assessment in future (in fact this was stated in the last paragraph of the discussion). Similarly, we have not studied any consumer preferences, but simply measured the serving temperature at food outlets. Nevertheless, in the previous round of review, we have added the additional references on consumer preference suggested by R2 to the manuscript with that perspective in mind (not altering the flow too much) (see previous additions in lines 113-116). To accommodate the reviewer, we have revised the statement about preference in the discussion section around lines 391-393.
Finally, we cannot completely understand the comment that we are not concerned with the readers. If you read the manuscript, you will notice that most readers, including those outside this particular field of research, are able to understand perfectly well what we did in our study. Further discussion on "preferred temperatures with temperatures causing harm" is outside the focus of the study. The study is fully justified (high per capita consumption of coffee in Brazil; hazard of very hot beverages in light of IARC classification; lack of such studies in Brazil), was correctly executed, has a clear aim and, last but not least, it is not pretentious. Of course the study may have certain gaps (such as studying consumer preference), but the authors do not believe it possible that a single publication may fill every and each one of all possible research gaps.